# Iterative Deep Compression : Compressing Deep Networks for Classification and Semantic Segmentation

## Abstract

Machine learning and in particular deep learning approaches have outperformed many traditional techniques in accomplishing complex tasks such as image classification (Krizhevsky et al., 2012), natural language processing or speech recognition (Hinton et al., 2012). Most of the state-of-the art deep networks have complex architecture and use a vast number of parameters to reach this superior performance. Though these networks use a large number of learnable parameters, those parameters present significant redundancy (de Freitas, 2013). Therefore, it is possible to compress the network without much affecting its accuracy by eliminating those redundant and unimportant parameters. In this work, we propose a three stage compression pipeline, which consists of pruning, weight sharing and quantization to compress deep neural networks. Our novel pruning technique combines magnitude based ones with dense sparse dense Han et al. (2016) ideas and iteratively finds for each layer its achievable sparsity instead of selecting a single threshold for the whole network. Unlike previous works, where compression is only applied on networks performing classification, we evaluate and perform compression on networks for classification as well as semantic segmentation, which is greatly useful for understanding scenes in autonomous driving. We tested our method on LeNet-5 and FCNs, performing classification and semantic segmentation, respectively. With LeNet-5 on MNIST, pruning reduces the number of parameters by 15.3 times and storage requirement from 1.7 MB to 0.006 MB with accuracy loss of 0.03%. With FCN8 on Cityscapes, we decrease the number of parameters by 8 times and reduce the storage requirement from 537.47 MB to 18.23 MB with class-wise intersection-over-union (IoU) loss of 4.93% on the validation data.

## 1 Introduction

Deep learning found its importance in different domains to solve tasks ranging from small-scale to large-scale problems. It has remarkably achieved human-level performance in image recognition tasks (He et al., 2016). Existing deep neural networks are very powerful but they require considerable storage and memory bandwidth. For example, AlexNet by Krizhevsky et al. (2012) has 61 million parameters, which is over 100 times more than LeCun et al. (1989) conventional model LeNet - 5 (LeCun et al., 1998). More parameters require more storage space and more computation. This makes it difficult to deploy deep neural networks on embedded devices and mobile platforms performing real time processing with limited memory and processing units. Still, not all weights in a network are important and there is in fact high redundancy in these parameters (Guo et al., 2016). Choosing the right and important parameters is essential to do the optimization between the network efficiency and resources used with minimal accuracy loss. While some success has been achieved in compressing deep neural networks performing classification, it has not been studied for networks performing semantic segmentation, where each pixel in the image is classified to a category making it possible to recognize multiple objects in an image. Semantic segmentation is extremely relevant for the self driving domain, where scenes have to be understood in real time, and is an important target for compression as deep learning methods have achieved significant success on it (Wu et al., 2016).

To achieve this goal, we present "iterative deep compression": a three stage pipeline as illustrated in figure 1, similar to "deep compression" proposed by Han et al. (2015). As is the case for many pipelines in this field, first we prune the number of parameters in the network by removing redundant and unimportant connections. To the remaining connections, we apply weight sharing so that the same weight can be shared by multiple connections across the network. Thus, only the shared weights and the indices mapping each remaining weight to its shared weight need to be stored. Finally, we apply quantization to reduce the number of bits required to store these shared weights.

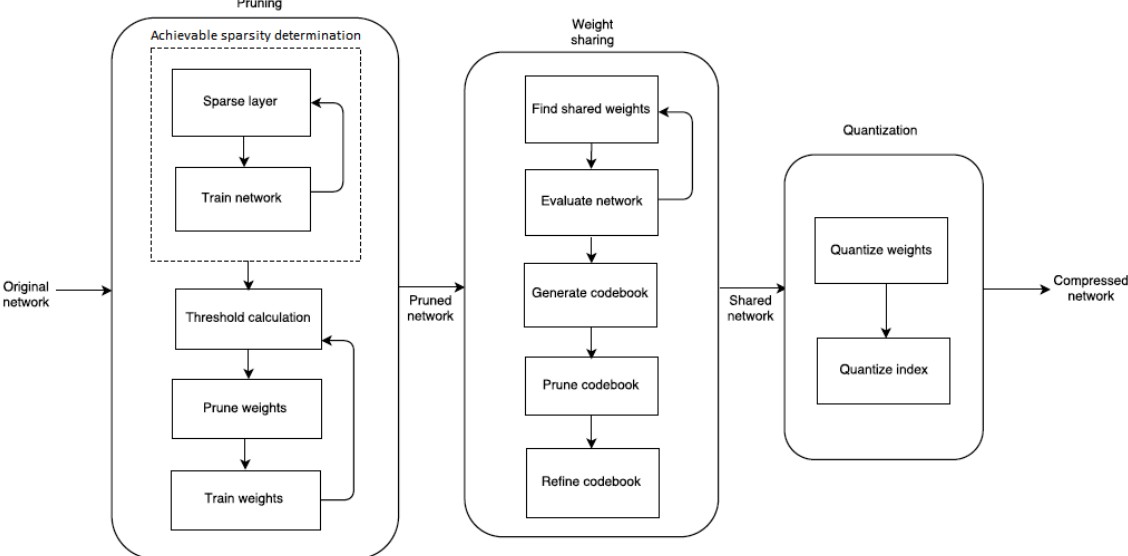

Figure 1: Three stage pipeline: pruning, weight sharing and quantization.

Our major contribution is the development of a pruning method that combines magnitude pruning with ideas from dense sparse dense (Han et al., 2016). That is done by iteratively finding for each layer its achievable sparsity instead of selecting a single threshold for the whole network and then applying dense sparse retraining while pruning. Moreover, we also investigate several options for weight sharing, such as sharing weights within layer and across the layers, examining the impact of each on network performance. We also explore different clustering techniques comparing their optimal number of clusters and resulting network performance.

In this work we compressed two state-of-the-art deep neural network architectures. The first performs classification on MNIST dataset (LeCun et al., 1998) while the second does semantic segmentation on Cityscapes (Cordts et al., 2016). While in some stages of our pipeline we may explore more than one technique, we always select the one performing best for our next stage.

## 2 RELATED WORK

In order to compress deep networks, a variety of methods have been proposed. Chen et al. (2015) accomplish network compression by HashNets, where they group the parameters of the network into hash buckets. These parameters are tuned with standard backpropagation during training, however, their binning is pre-determined by a hash function. They exploit the inherent redundancy in neural networks to achieve drastic reductions in model sizes. Gong et al. (2014) proposed another way to compress deep networks by vector quantization and found that the product quantization gives a good balance between model size and accuracy. They achieve 16 - 24 times compression on the state-of-the art CNN by classification accuracy loss of 1%. But both the compression methods consider only fully connected layers in the network. Lin et al. (2013) attempted to reduce the number of parameters by global average pooling the feature maps from the last convolutional layer of the network. Hubara et al. (2016) recently proposed binarized neural networks, where they use binary weights and activations during training. This bit wise operations substantially improve power-efficiency, but on the cost of accuracy loss.

Our work is based on the idea of network pruning. It helps to reduce both network complexity and over fitting (Hubara et al., 2016). Hassibi & Stork (1993) perform network pruning by using the information from second derivatives of the loss function and propose that such pruning is more accurate than magnitude based pruning, where smaller magnitude weights are eliminated. Their method does not require retraining the network after pruning, however, it is computationally expensive for large networks. Han et al. (2015) proposed magnitude based pruning while simultaneously retraining the network. They significantly reduce the number of parameters in the network without much impacting the accuracy, however, there is always the risk of deleting the important parameters. Guo et al. (2016) incorporate connection splicing in order to avoid incorrect pruning. In their splicing operation they enable the recovery of pruned connections if they are found important at any time, but they do not account for correlation between weights (Yang et al., 2016b). Recently, Han et al. (2015) proposed a deep compression pipeline. First they prune the unimportant connections based on their magnitude, then perform quantization to enforce weight sharing and finally apply Huffman encoding for lossless data compression. Their experiment on AlexNet reduced the number of parameters by 9x without loss of accuracy. Because of their significant effectiveness, our work is partly inspired by their work. However, we contribute to this work by improving the weight sharing technique and introducing dynamic threshold calculation for pruning. We also contribute to the work by Han et al. (2016) on dense sparse dense training by calculating the achievable sparsity in each layer.

## 3 Method

In this section we present our compression pipeline also depicted in figure 1, with its *Pruning*, *Weight Sharing* and *Quantization* steps.

**Pruning, From Dense to Sparse:**   The objective of the pruning technique is to identify and remove unimportant and redundant weights that would least affect the performance of the network. For this, we use the simple heuristic of quantifying the importance of weights using their absolute values (Han et al., 2015). In other words, the bigger the absolute weight value, the more important it is. But it is crucial to select one threshold weight value below which all the weights could be considered unimportant. We start with a pre-trained fully connected network. One approach is to choose a single threshold value for the entire network. However, this does not take into account the difference in the weight distribution across the layers. For example, as depicted in figure 2, the weight distribution of two different dense layers of the LeNet architecture differs. Therefore, we select the threshold value layerwise which means a different threshold value for each layer. To select the threshold values we introduced a hyper parameter to the network, the *sparsity*. This is the starting sparsity, initialized to 100% for each layer. It gives the percentage of weights of the network that are pruned, that is, set to 0 (Han et al., 2016). We determine the sparsity for each layer using algorithm 1 . We start from output to first convolutional layer. This process gives the approximate value of achievable sparsity in each layer without much impacting the accuracy of the network.

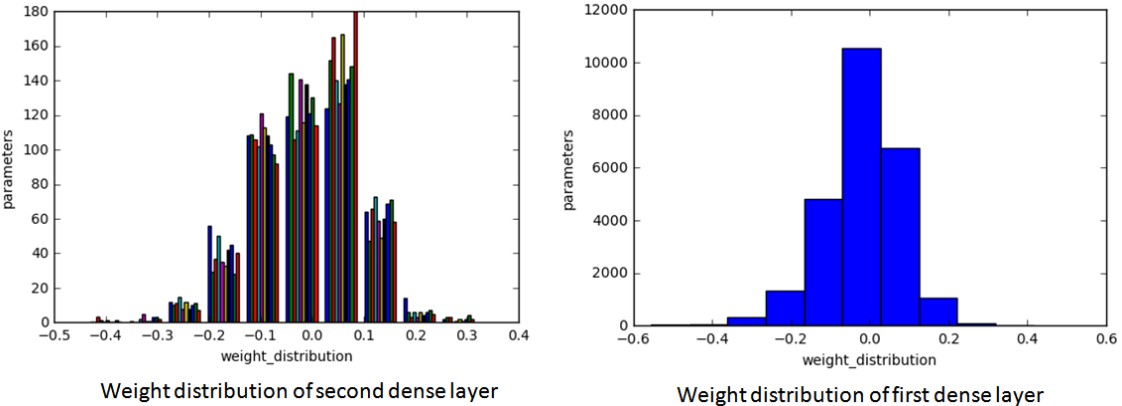

Weight distribution of second dense layer          Weight distribution of first dense layer

Figure 2: Weight distribution of two different dense layers of LeNet architecture.

---

**Algorithm 1:** Determine achievable sparsity in a layer

---

**Input:** error_tolerance, $Starting\_Sparsity_L$ for each layer defaults to 100, pre-trained network

**Output:** Achieved sparsity in each layer

**1** **foreach** *trainable layer L from n to* 1 **do**
**2** $\quad$ $Sparsity_L \leftarrow Starting\_Sparsity_L$ ;
**3** $\quad$ Prune the weights in layer $L$ up to defined sparsity $Sparsity_L$ ;
**4** $\quad$ $t \leftarrow 0$ ;
**5** $\quad$ $current\_error_t \leftarrow$ computed error with the modified network ;
**6** $\quad$ **while** $current\_error_t > error\_tolerance$ **do**
**7** $\quad\quad$ Train the model ;
**8** $\quad\quad$ Prune again up to same level, $Sparsity_L$ ;
**9** $\quad\quad$ $t \leftarrow t + 1$ ;
**10** $\quad\quad$ $current\_error_t \leftarrow$ current error of the network ;
**11** $\quad\quad$ **if** $current\_error_t < current\_error_{t-1}$ **then**
**12** $\quad\quad\quad$ Decrease the sparsity, $Sparsity_L$, by 1% ;
**13** $\quad\quad$ **end**
**14** $\quad$ **end**
**15** $\quad$ $Achieved\_Sparsity_L \leftarrow Sparsity_L$ ;
**16** **end**
**17** **return** $Achieved\_Sparsity$

---

In this algorithm 1, first, we initialize the sparsity for each layer and receive as input the pre-trained network and a parameter called 'error tolerance' for the network. Error tolerance is computed by adding a very small degradation tolerance to baseline error of the model, where baseline error is the error of the pre-trained network before pruning. Then, we initialize sparsity to 100% for a layer, that is, setting all its weights to zero and then we evaluate error of the network as depicted in step 3 and 5. In step 6, we compare this error with the defined error tolerance of the network. If the current error is more than the defined error tolerance, we train the network and again prune it at same sparsity level. We repeat this process until the error of the network converges at that sparsity level and it stops improving by further retraining. We decrease the sparsity in the layer by 1% after every such convergence, as shown in step 12.

Once we got the approximated sparsity level for each layer, we start pruning our network, depicted in algorithm 2. The achieved sparsity for each layer and the overall error tolerance of the network are received as parameters from the last step. We also receive the original pre-trained weights and an extra sparsity for each layer. This extra sparsity will be added to the starting sparsity found in the previous algorithm. Then we calculate the threshold value for each layer $\lambda_L$ that is needed to make the layer $Sparsity_L$ sparse, pruning the weights of each layer. However, mistakenly pruning important connections or over pruning could cause high accuracy loss (Guo et al., 2016). In order to compensate the unexpected loss, we retrain the network and enable the connection recovery, that is, a pruned value is not henceforth always zero as is done in Han et al. (2015), but can regain a positive absolute value after retraining. Finding important connections in a certain network is also extremely difficult, therefore, we conduct pruning and training iteratively and continually maintain the old weights from the previous iteration. After each iteration, a pruning threshold is selected from trained weights of the previous iteration. Hence, we are dynamically calculating the threshold for each layer. After pruning, we evaluate the pruned model and compare this current error with the defined error tolerance of the network. If the current error is greater than the error tolerance, we retrain the network and prune it again as can be seen in step 11 and 12. All the layers of the network are pruned and trained together because of the dependence of each layer to its previous layer. We keep track of the error from every iteration and we stop retraining if the network converges, that is, error is not improving any more at defined sparsity level. This error has been compared with the error tolerance of the network as depicted in step 15. We decrease the sparsity by 1% and again start the iterative process pruning and retraining if the current error is still less than the error tolerance. Otherwise, it returns pruned network as can be seen in step 19. This sums up pruning of the network where we iteratively undergo the dense sparse phase under the constraint of sparsity and error tolerance.

---

**Algorithm 2:** Dynamic threshold calculation and pruning of the network

---

**Input:** error_tolerance, $Achieved\_Sparsity$ for each layer, $Extra\_Sparsity_L$ for each layer, pre-trained network

**Output:** Sparse network

1   $Sparsity_L \leftarrow Achieved\_Sparsity_L + Extra\_Sparsity_L$ ;
2   **Sparse phase(for each layer simultaneously):**
3   Get weights $W_L$ for each layer separately ;
4   Calculate threshold, $\lambda_L$ needed to obtain $Sparsity_L$ ;
5   Prune $i_{th}$ weight, $W_L^{ith} \leftarrow 0$ if $W_L^{ith} < \lambda_L$ ;
6   **Dense phase** ;
7   $t \leftarrow 0$ ;
8   $current\_error_t \leftarrow$ Evaluate the network ;
9   **do**
10    **do**
11     Retrain the network ;
12     Again prune the network using, $\lambda_L$ ;
13     $current\_error_t \leftarrow$ Evaluate the network ;
14     $t \leftarrow t + 1$ ;
15    **while** $current\_error_{t-1} > current\_error_t$;
16    Decrease the $Sparsity_L$ in each layer by 1% ;
17    Recalculate $\lambda_L$ ;
18 **while** $current\_error > error\_tolerance$;
19 **return** *pruned network*

---

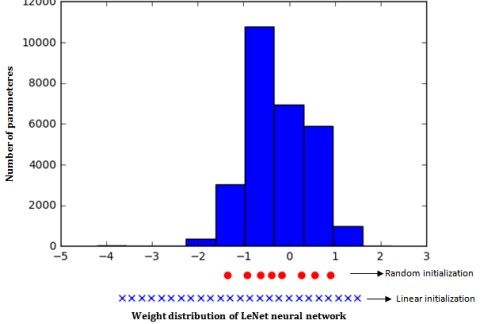

Figure 3: Initial cluster centers by random and linear centroid initializations

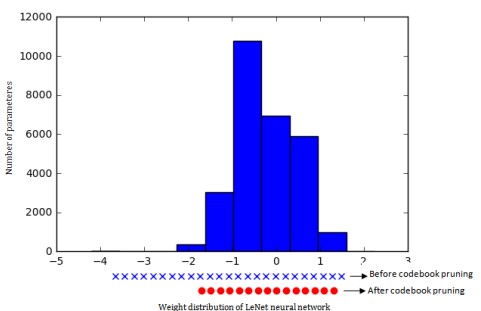

Figure 4: Linear initialization of clusters centers before and after codebook pruning

**Weight Sharing**   limits the number of effective weights to store by finding the weight values that could be shared across multiple connections without affecting the accuracy of the network. Weights can be shared, first, only within a layer, that is finding shared weights among the multiple connections within a layer and second, across all the layers, that is, finding shared weights among multiple connections across all the layers. This can be done using clustering. The idea is that all the weights that belong to one cluster share the same weight value, partitioning the n original weights into k clusters such that $n >> k$. The performance of the network depends upon the quality of the clustering algorithm. The value of the cluster's centroid is assigned to all the weight values within that cluster. So at the end we need to store only the centroid values to represent the weight values of the network instead of storing all the weight values individually. We examine two different clustering algorithms: mean shift clustering and k-means. K-means is very sensitive to the initial position of the cluster centers (Celebi et al., 2013), so we examine two different initialization methods: random and linear initialization (Han et al., 2015). The stepwise algorithm for k-means with linear initialization is illustrated in algorithm 3.

**Refinement of the shared weights**   As depicted in figure 3, linear initialization results in cluster centers scattered over the entire range. However, cluster centers at the extreme ends of the distri-

---

**Algorithm 3:** Procedure used to find cluster centers with linear initialization

---

**Output:** cluster_centers
**Input:** starting_number_of_partitions, error_tolerance, all_weights_of_network, step_size
1  range ← |max $all\_weights\_of\_network$| + |min $all\_weights\_of\_network$| ;
2  number_of_partitions ← starting_number_of_partitions ;
3  **while** $current\_error > error\_tolerance$ **do**
4      number of clusters = number_of_partitions +1 ;
5      cluster_centers = points dividing range equally to n partitions ;
6      Perform k-means with these cluster_centers initialization ;
7      Replace all the weight values of the network with their nearest cluster center ;
8      Calculate current_error by evaluating the model ;
9      number_of_partitions ← number_of_partitions + step_size ;
10 **end**
11 **return** $cluster\_centers$

---

bution represent few original weights. It could be the case, however, that those underrepresented weight values are really important to the network's performance. Thus we will evaluate the effects of merging them with the other shared weights, by pruning the code book.

**Pruning of the codebook**  To check the possibility of reducing down the number of shared weights, we first examine the number of weights mapped to each center in the codebook. We found that there are some codes that are being assigned to zero or very few weight values. So we prune such codes. For this pruning, we empirically chose a fixed threshold of 25. Therefore, any code which is assigned to less than 25 weight values would be pruned and removed from the codebook. All the weight values assigned to such codes/cluster centers would be reassigned to next closest cluster center. In our experiments we have observed that this does not impact much the accuracy of the network. The cluster centers before and after codebook pruning are shown in figure 4.

**Fine tuning of the codebook**  To fine tune the cluster centers we use the gradient approach (Han et al., 2015). First, gradients for each weight value are calculated using Theano symbolic differentiation. Then, the gradients of the weight values that belong to one cluster are grouped together. All the calculated gradients are grouped according to the cluster they belong and summed to give one value per cluster. This value is then multiplied with the learning rate and subtracted from the cluster centers obtained from the previous step. These new values are called fine-tuned cluster centers. Now these fine-tuned values are used as shared weights for the network.

**Quantization**  Quantization is performed to reduce the number of bits required to represent each shared weight value. We use fixed point quantization to convert floating point weight values to fixed point weight values. Fixed point implementation facilitates the potential deployment on embedded systems (Lin et al., 2016). Based on state-of-the-art by Han et al. (2015) we convert the 32-bit floating point weight values to 8-bit fixed point values for each fully convolutional layer and to 5-bit fixed point values for each fully connected layer. The conversion of floating point to fixed point is done using the following formulation, where $q_{format}$ is the user defined bit width (Lin et al., 2016):

$$\text{Fixed point value} = \text{Floating point value} * 2^{q_{format}} \tag{1}$$

## 4  EXPERIMENTS AND RESULTS

We experimentally analyzed the three stage pipeline discussed above and applied it to some popular pre-trained networks. In section 4.1 and 4.2 we present our experiment results for LeNet-5 on MNIST and FCN8 on cityscapes respectively.

### 4.1  RESULTS FOR LENET-5 ON MNIST

We trained LeNet on the training set of MNIST to accomplish the classification task. The optimizer Adam was used to train the network. We trained the network for 15 epochs with learning rate of

$10^{-3}$. Our trained LeNet has achieved the accuracy of 99.30% on the validation set of MNIST. So, we have 99.30% as baseline accuracy and consequently 0.70% as baseline error rate of the network.

**Pruning**    The first step was to select the threshold used to prune the network. Table 1 gives the comparison between our two threshold selection methods. In the first, we fixed one threshold value for all layers while in the second we dynamically calculate a different threshold for each layer. Dynamic threshold selection achieved substantially better results.

Table 1: Accuracy based comparison of the two different threshold selection methods

| Threshold selection methods | Achievable Sparsity (% of Zero Weights) |
|---|---|
| One fixed threshold | 86.30 % |
| Different dynamically selected thresholds | 93.47 % |

While initializing the sparsity levels for each layer, we increased the above mentioned percentage by 5 to 6% following our algorithm. Based on these defined sparsity levels, the initial threshold has been calculated for each layer. We prune each layer of the network with these threshold values. A plot of the weight distribution of second convolutional layer before and after first pruning is shown in figure 5.

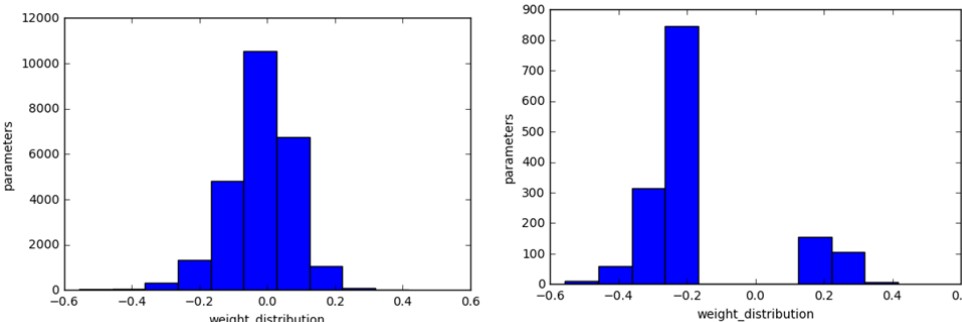

Figure 5: Weight distribution of second convolutional layer of LeNet before pruning (in the left) and after pruning (in the right)

We iteratively perform pruning and retraining using the same optimizer and learning rate on all network weights until the current validation error of the network becomes less than the defined error tolerance. Our pruning runs for 42 iterations in total for the whole network. Figure 6 depicts the convergence of error rate and accuracy rate after each iteration at initialized sparsity level. As it can be seen, after a few iterations the accuracy of the network is not improving anymore so we decreased the sparsity level by 1% in each layer and again start iteratively pruning and retraining at new decreased sparsity levels. In table 2 we compare layer by layer pruning results between our method and Han et al. (2015).

**Weight Sharing**    Next, we applied weight sharing techniques on these remaining weights of the network. We perform clustering on all the non-zero weights. Weight sharing further reduced the number of weights to be stored, as depicted in table 3, where the comparison of each weight sharing technique we explored is shown in terms of number of clusters and accuracy achieved.

So k-means clustering with linear initialization across all the layers gives the best results. We fine tune these cluster centers by gradient approach discussed in section 3. The accuracy of the network remained unchanged after fine tuning.

**Quantization**    Next, we applied quantization to reduce the number of bits required to store the tuned cluster centers. We applied 8-bit fixed point quantization for convolutional layers and 5-bit fixed point quantization for fully connected layers. Table 4 gives the accuracy statistics and reduction in size after each stage of pipeline.

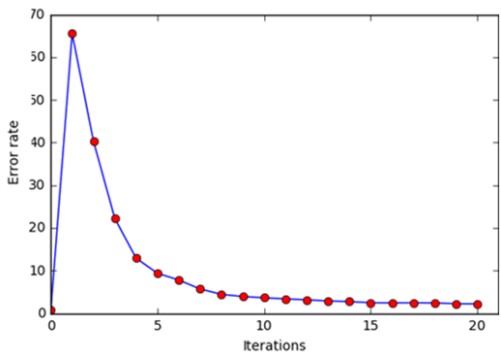 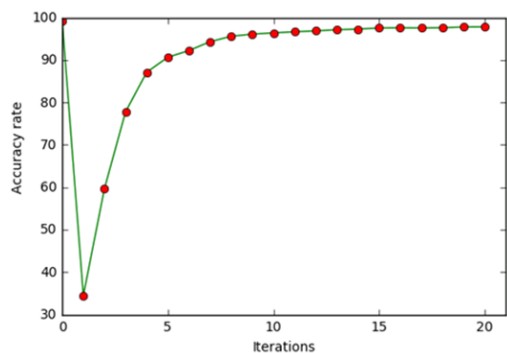

(a) Convergence of error rate after each iteration of pruning and training

(b) Convergence of accuracy rate after each iteration of pruning and training

Figure 6: Convergence of error rate (a) and accuracy rate (b) after each iteration of pruning and training

Table 2: Comparison of our pruning results on LeNet-5 with that of Han et al. (2015)

| Model | Layer | Percentage of remaining parameters [ours] | Number of remaining parameters [ours] | Percentage of remaining parameters (Han et al., 2015) | Number of remaining parameters (Han et al., 2015) |
|---|---|---|---|---|---|
| LeNet-5 | F2 | 5% | 250 | 19 % | 950 |
| | F1 | 6% | 24000 | 8 % | 32000 |
| | C2 | 15% | 3750 | 12 % | 3000 |
| | C1 | 50% | 90 | 66 % | 330 |
| Total | | $\approx 6.5\%$ | 28090 | $\approx 8\%$ | 36280 |
| Accuracy | | 99.27% | | 99.26 % | |
| Storage Requirement | | 6 KB | | 44 KB | |

## 4.2 RESULTS FOR FCN ON CITYSCAPES

We conducted second experiment on FCN (Yang et al., 2016a) performing semantic segmentation task on the Cityscapes dataset. Our trained FCN8 has achieved the baseline IU of 64.75% and baseline error rate of 35.25% on the validation set of cityscapes.

**Pruning** In this experiment, dynamic threshold calculation is used to calculate the threshold as it is evident from table 1 that dynamic threshold selection achieved substantially better results. We iteratively perform the pruning and retraining operation with the same optimizer and learning rate until the current validation error of the network becomes less than the defined error tolerance. It

Table 3: Comparison of all the weight sharing techniques discussed above

| Weight sharing technique | Number of clusters found | Accuracy achieved |
|---|---|---|
| Mean shift | 12 | 99.05% |
| k-means with random initialization | 8 | 98.94% |
| k-means with linear initialization within layers | 24 | 99.14% |
| k-means with linear initialization across all the layers | 15 | 99.27% |

Table 4: Accuracy statistics and reduction in size after each stage of pipeline

| Stages of pipleline | Storage requirement of parameters (in MB) | Reduction in storage requirement (in %) | Accuracy |
|---|---|---|---|
| Baseline | 1.7 MB | - | 99.30% |
| Pruning | 0.11 MB | 93.52% | 99.26% |
| Pruning + Weight sharing | 0.008 MB | 99.50% | 99.28% |
| Pruning + Weight sharing + Pruned code book | 0.007 MB | 99.58% | 99.27% |
| Pruning + Weight sharing + Pruned codebook + Quantization | 0.006 MB | 99.59% | 99.27% |

runs for 32 iterations for the whole network. Figure 7 depicts the convergence of error rate and accuracy rate after each iteration at initialized sparsity level. Plot of the weight distribution of first convolutional layer before and after first pruning is shown in figure 8. Table 5 depicts the compression statistics after pruning and table 6 depicts the mean IU statistics and reduction in storage after pruning.

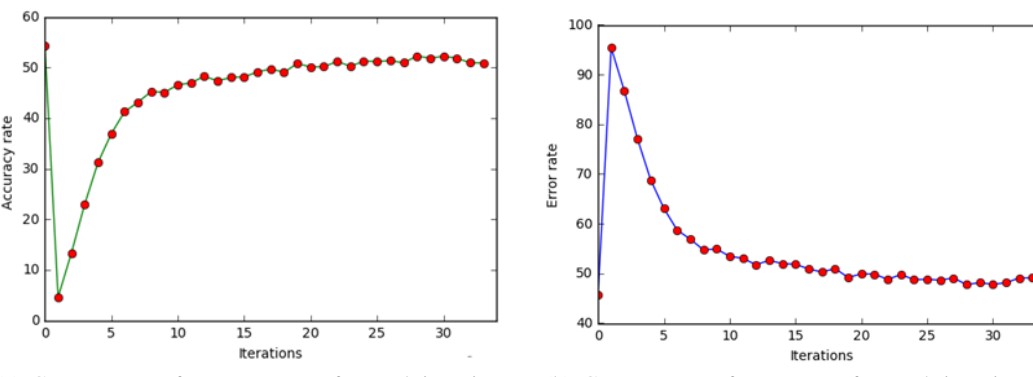

(a) Convergence of accuracy rate after each iteration of pruning and training

(b) Convergence of error rate after each iteration of pruning and training

Figure 7: Convergence of accuracy rate (a) and error rate (b) after each iteration of pruning and training

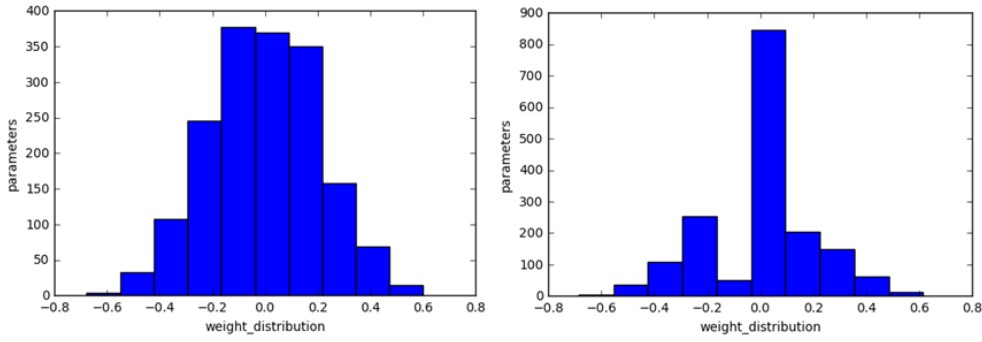

Figure 8: Weight distribution of first convolutional layer of FCN before pruning (in the left) and after pruning (in the right)

**Weight sharing**    To find the shared weights, we applied k-means clustering with linear initialization as it achieved substantially better results in the previous experiment. To find the optimum number of clusters, we evaluate the number of clusters ranging from 10 to 1200. Figure 9 depicts the achieved IOU score corresponding to number of clusters found in this range. It can been seen

Table 5: FCN compression results after pruning

| Layers of the network to be pruned | Total parameters before pruning | Achieved sparsity in each layer | Total non-zero parameters after pruning | Remaining non-zero weights after pruning in % (P) |
|---|---|---|---|---|
| C1 | 1792 | 49 % | 913 | 51 % |
| C2 | 36928 | 84 % | 5908 | 16 % |
| C3 | 73856 | 69 % | 22895 | 31 % |
| C4 | 147584 | 69 % | 45751 | 31 % |
| C5 | 295168 | 69 % | 91502 | 31 % |
| C6 | 590080 | 70 % | 177024 | 30 % |
| C7 | 590080 | 69 % | 182924 | 31 % |
| C8 | 1180160 | 79 % | 247833 | 21 % |
| C9 | 2359808 | 81 % | 448363 | 19 % |
| C10 | 2359808 | 84 % | 377569 | 16 % |
| C11 | 2359808 | 83 % | 401167 | 17 % |
| C12 | 2359808 | 83 % | 401167 | 17 % |
| C13 | 2359808 | 83 % | 401167 | 17 % |
| C14 | 102764544 | 89 % | 11304099 | 11 % |
| C15 | 16781312 | 89 % | 1845944 | 11 % |
| C16 | 77843 | 89 % | 8562 | 11 % |
| C17 | 9747 | 89 % | 1072 | 11 % |
| C 18 | 3268 | 89 % | 359 | 11 % |
| C19 | 4883 | 89 % | 537 | 11 % |
| C20 | 3268 | 89 % | 359 | 11 % |
| C21 | 3268 | 89 % | 359 | 11 % |
| C22 | 3268 | 90 % | 326 | 10 % |
| C23 | 3268 | 80 % | 653 | 20 % |
| Total sparsity | 134369357 | 86.44 % | 18233102 | 13.56 % |

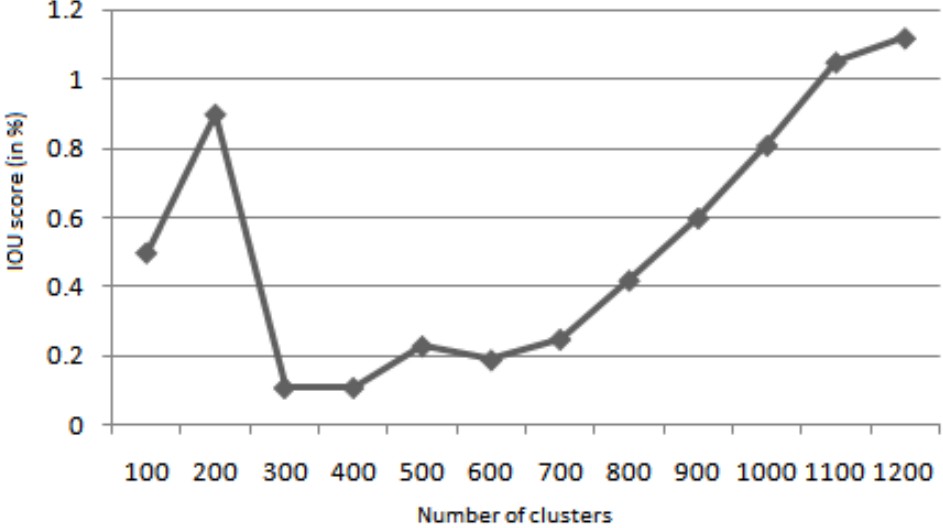

Figure 9: Number of clusters vs IOU score

that as the number of clusters increases the IOU score is improving, however, the improvement is very slow. Due to time constraints, we did not continue with weight sharing and directly applied quantization to the non-zero weights after pruning.

**Quantization**    We applied 8 bit quantization to pruned weights. The size of the network reduced from 537.47 MB to 18.23 MB as depicted in table 6.

Table 6: Reduction in storage requirement after pruning and quantization

| Stages of pipleline | Storage requirement of parameters (in MB) | Reduction in storage requirement (in %) | IOU |
|---|---|---|---|
| Baseline | 537.47 MB | - | 64.75 % |
| Pruning | 72.93 MB | 86.43% | 61.25 % |
| Pruning + Quantization | 18.23 MB | 96.60% | 59.82 % |

## 5    CONCLUSION AND FUTURE WORK

Deep learning approaches have demonstrated that they can outperform many traditional techniques, but because of their complex architecture in terms of more stacked layers and a large number of parameters, it is challenging to deploy these deep networks on mobile devices with limited hardware requiting real time predictions. This work contributes to the previous research on compression of deep networks performing classification. Moreover, we have also presented the compression of a network that performs semantic segmentation. We implemented a three stage deep compression pipeline of pruning, weight sharing, and quantization. Using different sparsity levels, we calculate different thresholds in each layer to perform pruning. Our "layerwise threshold" initialization method has shown promise in providing a good trade-off between sparsity and network performance. We also examined two different weight sharing possibilities: finding shared weights within a layer or across all the layers of the network. We extend the previous work by Han et al. (2015) with k-means using linear initialization by merging underrepresented shared weights. Finally, we quantize these shared weights, based on state-of-the-art by Han et al. (2015), in which they used fixed point quantization. The experimental results show that our method compresses the number of parameters in LeNet - 5 and FCN by 15.3x and 8x, respectively. We reduce the storage requirement for LeNet from 1.7 MB to 0.007 MB and for FCN from 547 MB to 18.23 MB. The reduction in storage for the FCN has extended network compression to more sophisticated tasks such as object detection and segmentation.

In our work, iterative pruning is performed to get rid of unimportant connections. This process takes 35 hours for LeNet with 430180 parameters on MNIST and 42 days for FCN with approximately 134M parameters on Cityscape to reach an optimum level of sparsity and performance. While not prohibitive, this process could be sped up by different approaches such as masking by Guo et al. (2016), where they have a gradient based heuristic for determining which weights do not come back. It would also be interesting to carry out the experiment on bigger networks and datasets for classification. Also recently, smaller deep neural network architectures, such as SqueezeNet, by Han et al. achieved AlexNet-level accuracy on ImageNet with 50x fewer parameters and is thus feasible to be deployed on FPGAs and other hardware with limited memory (Iandola et al., 2016). Compressing already smaller deep neural networks could further reduce the number of parameters and make them even more efficient on embedded systems.

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
