# OpenReview forum: "Iterative Deep Compression : Compressing Deep Networks for Classification and Semantic Segmentation"
_ICLR.cc/2018/Conference — Reject_

### Official Review · AnonReviewer3 · 2017-11-26
**Pruning NNs with layer-dependent parametrization, combined with other approaches for space reduction (weight sharing, quantization) significantly decrease space requirements without hurting accuracy. However, this comes at the cost of significant pruning time.**

**Rating:** 6
**Confidence:** 4

**Review:**

quality: this paper is of good quality
clarity: this paper is very clear
originality: this paper combines original ideas with existing approaches for pruning to obtain dramatic space reduction in NN parameters.
significance: this paper seems significant.

PROS
- a new approach to sparsifying that considers different thresholds for each layer
- a systematic, empirical method to obtain optimal sparsity levels for a given neural network on a task.
- Very interesting and extensive experiments that validate the reasoning behind the described approach, with a detailed analysis of each step of the algorithm.

CONS
- Pruning time. Although the authors argue that the pruning algorithm is not prohibitive, I would argue that >1 month to prune LeNet-5 for MNIST is certainly daunting in many settings. It would benefit the experimental section to use another dataset than MNIST (e.g. CIFAR-10) for the image recognition experiment.
- It is unclear whether this approach will always work well; for some neural nets, the currently used sparsification method (thresholding) may not perform well, leading to very little final sparsification to maintain good performance.
- The search for the optimal sparsity in each level seems akin to a brute-force search. Although possibly inevitable, it would be valuable to discuss whether or not this approach can be refined.

Main questions
- You mention removing "unimportant and redundant weights" in the pruning step; in this case, do unimportant and redundant have the same meaning (smaller than a given threshold), or does redundancy have another meaning (e.g. (Mariet, Sra, 2016))?
- Algorithm 1 finds the best sparsity for a given layer that maintains a certain accuracy. Have you tried using a binary search for the best sparsity instead of simply decreasing the sparsity by 1% at each step? If there is a simple correlation between sparsity and accuracy, that might be faster; if there isn't (which would be believable given the complexity of neural nets), it would be valuable to confirm this with an experiment.
- Have you tried other pruning methods than thresholding to decide on the optimal sparsity in each layer?
- Could you please report the final accuracy of both models in Table 2?

Nitpicks:
- paragraph break in page 4 would be helpful.

---

> ### Author Response · Authors · 2017-12-15
> **Answers to above all mentioned points:**
>
> 1) We removed unimportant weights (smaller than a given threshold). Or, we can say both have the same meaning.
>
> 2) It is really an interesting idea and might perform faster, however, considering the complexity of the network, there might be convergence problem as we would change the sparsity abruptly. Indeed, we could try it out with an experiment.
>
> 3) We used the simple heuristic of quantifying the importance of weights using their absolute values. We could try the other ways in future work.
>
> 4) We will add it in our next version.
>
> 5) We will take this in our next version.

---

> ### Author Response · Authors · 2017-12-26
> **Pruning time:**
>
> The number of iterations mentioned for pruning is for the whole network and most of the compression happens in the first 10 iterations, meaning that the method is not so time-consuming as it may seem from the total runtime reported.

---

### Official Review · AnonReviewer1 · 2017-11-27
**iterative deep compression**

**Rating:** 4
**Confidence:** 4

**Review:**

The paper presents a method for iteratively pruning redundant weights in deep networks. The method is primarily based on a 3-step pipeline to achieve this objective. These three steps consist of pruning, weight sharing and quantization. The authors demonstrate reduction in model size and number of parameters significantly while only undergoing minor decrease in accuracy.

Some of the main points of concern are below :

 - Computational complexity - The proposed method of iterative pruning seems quite computationally expensive. In the conclusion, it is mentioned that it takes 35 days of training for MNIST. This seems extremely high, and given this, it is unclear if there is much benefit in further reduction in model sizes and parameters (by the proposed method) than those obtained by existing method such as Han etal.

 - The novelty in the paper is quite limited and is mainly based on combining existing methods for pruning, weight sharing and quantization. The main difference from existing method seems to be the inclusion of layerwise threshold for weight pruning instead of using a single global threshold.

 - The results shown in Table 2 do not indicate much difference in terms of number of parameters between the proposed method and that of Han etal. For instance, the number of overall remaining parameters is 6.5% for the proposed method versus 8% for Deep Compression. As a result, the impact of the proposed method seems quite limited.

 - The paper in the title and abstract refers to segmentation as the main area of focus. However, there does not seem to be much related to it except an experiment on the CityScapes dataset.

---

> ### Author Response · Authors · 2017-12-14
> **Answers to points 1, 2, 3 and 4**
>
> 1)  It’s a typo. It took 35 hours for MNIST. We will correct it in our next revision.
>
> 2)  Following points highlights the differences between the existing and our approach.
>
>      -	  We evaluate different threshold initialization methods for  weight pruning. To determine those thresholds, we
>          conducted an experiment in which we calculate the minimum achievable sparsity in each layer.
>
>      -	  We explore different clustering techniques to find shared weights. We examine the impact of density based
>           meanshift clustering and unsupervised k-means clustering with random and linear centroid initialization.
>
>      -	 We also evaluated different weight sharing possibilities. First, only within a layer, that is finding shared weights
>          among the multiple connections within a layer (Han et al.) and second, across all the layers, that is, finding
>          shared weights among multiple connections across all the layers. We found that the second method
>          outperforms the first one.
>
>      -	 We show the trade-off between the number of clusters by state-of-the-art weight sharing technique (k-means
>         clustering with linear centroid initialization) and network performance. We also proposed and implemented
>         ways to improve it.
>
>      -	  We compress and evaluate our method on a fully convolutional network performing semantic segmentation
>          and we are not aware of any state-of-the-art technique that obtains good compression rates for such networks.
>
> 3)  We successfully demonstrated the flexibility of our method by testing it on fully convolutional network performing other task than classification.
>
> We also outperformed the existing pruning method (Han et al.) not only in terms of compression statistics but also in accuracy results.
>
> However, a better comparison could be done with some other network / dataset, such as inception and ImageNet, but that the focus was indeed on the segmentation.
>
> 4) Currently, there is no date set that could adequately captures the complexity of real-world urban scenes [1]. Cityscapes is a benchmark suite and large-scale dataset to address the understanding of complex urban street scenes and there was no experiment performed on this very relevant dataset. So, we focus to use cityscapes high quality images in our experiments and address the problem of real time computation with limited hardware resources in autonomous driving
>
> Also, in this research work, one of our main goals was to perform compression on networks performing some other tasks than just classification. So, unlike all the previous works, where compression is only performed on networks performing classification, we evaluated and performed compression on networks for semantic segmentation.
>
> References: [1] Cordts, Marius, et al. "The cityscapes dataset for semantic urban scene understanding." Proceedings of the IEEE Conference on Computer Vision and Pattern Recognition. 2016.

---

### Official Review · AnonReviewer2 · 2017-11-28
**Good work but much similarity with an existing work**

**Rating:** 5
**Confidence:** 4

**Review:**

This paper inherits the framework proposed by Han[1]. A pruning, weight sharing, quantization pipeline is refined at each stage. At the pruning stage, by taking into account difference in the distribution across the layers, this paper propose a dynamic threshold pruning, which partially avoids mistakenly pruning important connections. As for the weight sharing stage, this paper explores several ways to initialize the clustering method. The introduction of error tolerance gives us more fine-grained control over the compression process.

Here are some issues to be paid attention to:

1. The overall pipeline including the last two stage looks quite similar to Han[1]. Though different initialization methods are tested in this paper, final conclusion does not change.

2. The dynamic threshold pruning seems to be very time-consuming. As indicated from the paper, only 42 iterations for MNIST and 32 iterations for Cityscapes are required. Whether these number works for each layer or total network should be clarified.

3. Fig 7(a) says it's error rate while it plots accuracy rate.

4. Experiments on popular network structure such as residual connection should be conducted, as they are widely used nowadays.


References:
[1] Song Han, Huizi Mao, and William J Dally. Deep compression: Compressing deep neural networks with pruning, trained quantization and huffman coding. arXiv preprint arXiv:1510.00149, 2015

---

> ### Author Response · Authors · 2017-12-13
> **Answer to point 1**
>
> Here, we would like to highlight the differences between the Han[1] and our approach for the weight sharing stage:
>
> Yes, we evaluated the different initialization methods. And, we also evaluated different weight sharing possibilities. First, only within a layer, that is finding shared weights among the multiple connections within a layer and second, across all the layers, that is, finding shared weights among multiple connections across all the layers. We found that the second method outperforms the first one in our case, however, Han[1] stated and used the first one. Comparison of weight sharing techniques discussed above:
>
>
> Weight sharing techniques for LeNet on Mnist	                  Number of clusters found	     Accuracy achieved
>
> k-means with linear initialization within layers [Han]	                       24	                                         99.14%
> k-means with linear initialization across all the layers [ours]	       25	                                         99.28%
>
> We further improved our k-means with linear initialization across all the layers by checking the possibility of reducing down the number of shared weights. For this, we added one more step to the pipeline, that is, pruning of the codebook. For LeNet on Mnist, we reduced the number of shared weights from 25 to 15 by applying codebook pruning with accuracy loss of just 0.01%. So, our approach gives the optimal trade-off between number of shared weights and loss of accuracy.

---

> ### Author Response · Authors · 2017-12-14
> **Answers to points 2, 3 and 4**
>
> 2) This is for the whole network. In each iteration, we performed pruning and retraining on each layer simultaneously. We will clarify this in the next version. Moreover, most of the compression happens in the first 10 iterations, meaning that the method is not so time-consuming as it may seem from the total runtime reported.
>
> 3) We will correct it in our next revision.
>
> 4) Yes, it would be really interesting to see how our compression works on residual connections. This could be our future research work.
>
> In this research work, one of our main goal was to perform compression on networks performing some other tasks than just classification. So, unlike all the previous works, where compression is only performed on networks performing classification, we also evaluated and performed compression on networks for semantic segmentation. In this work, we tried to address the problem of real time computation with limited hardware resources in autonomous driving.  Semantic segmentation is greatly useful for understanding scenes in autonomous driving. So, we tried to compress a network performing semantic segmentation on Cityscapes dataset.
>
> Also, we are not aware of any state-of-the-art technique that obtains good compression rates for fully convolutional networks, so we were interested to see how much compression could be achieved on a network without any fully connected layer. Thus, we decided to compress the fully convolutional network.

---

### Author Response · Authors · 2018-01-05
**Info update**

Changes are made to the paper.

---

### Decision · Program_Chairs · 2018-01-29
**ICLR 2018 Conference Acceptance Decision**

**Decision:**

Reject

**Comment:**

This paper presents a new pipeline for nn compression that extends that of Han et. al, but show that it reduces parameters further, maintains higher accuracy and can be applied to methods behind classification (semantic segmentation). While the authors found the paper clearly written, excepting for some typos, and potentially useful, there were questions about originality, and significance.

- Reviewers were not completely convinced the method was different enough from deep compression: "The overall pipeline including the last two stage looks quite similar to Han[1].", or that enough focus was paid to the differences inherent with classification focused work: "The paper in the title and abstract refers to segmentation as the main area of focus. However, there does not seem to be much related to it except an experiment on the CityScapes dataset."

- In terms of impact, the additional benefits from pruning seem to require a significant amount of computation, and the reviewers were not convinced these were worth a small gain in compression. Furthermore, authors felt that this approach was not being applied to the most state-of-the-art approaches to demonstrate their use.